# Ionic Liquids-Functionalized Zeolitic Imidazolate Framework for Carbon Dioxide Adsorption

**DOI:** 10.3390/ma12152361

**Published:** 2019-07-25

**Authors:** Xuyan Song, Jialin Yu, Min Wei, Ran Li, Xi Pan, Guoping Yang, Haolin Tang

**Affiliations:** 1Technology Centre of Hubei China Tobacco Industry Co., LTD., Wuhan 430051, China; 2State Key Laboratory of Advanced Technology for Materials Synthesis and Processing, Wuhan University of Technology, Wuhan 430070, China

**Keywords:** ionic liquid, zeolitic imidazolate framework, adsorption, carbon dioxide, micropores

## Abstract

Ionic-liquid-functionalized zeolitic imidazolate frameworks (ZIF) were synthesized using the co-ligands of 2-methylimidazole and amine-functionalized ionic liquid during the formation process of frameworks. The resulting ionic-liquid-modified ZIF had a specific surface area of 1707 m^2^·g^−1^ with an average pore size of about 1.53 nm. Benefiting from the large surface area and the high solubility of carbon dioxide in ionic-liquid moieties, the synthesized materials exhibited a carbon dioxide adsorption capacity of about 24.9 cm^3^·g^−1^, whereas it was 16.3 cm^3^·g^−1^ for pristine ZIF at 25 °C under 800 mmHg. The results demonstrate that the modification of porous materials with ionic liquids could be an effective way to fabricate solid sorbents for carbon dioxide adsorption.

## 1. Introduction

Emissions of carbon dioxide induced by anthropogenic activity are realized as one of the major contributions to the phenomenon of global warming and the associated environmental problems [1,2]. Capture and storage of carbon dioxide under mild conditions are, therefore, essential for the reduction of carbon emission. In addition, as a valuable feedstock in industry, the captured carbon dioxide can be further applied in various industrial areas [3]. Thus, development of advanced technology for carbon dioxide capture or adsorption is of great importance.

The often-applied technology for carbon dioxide capture in industry relies on alkaline amine-based solutions, which exhibit great adsorption capacity [4,5]. However, the high volatility, decomposition of amine-based compounds, and the high corrosion to scrubbers hinder their wide application [6,7,8]. Thus, solid sorbents attracted research attention due to their safety, good cyclability, high selectivity, and convenience for transport and storage [9,10]. As an example, amine-functionalized porous materials were realized as efficient solid sorbents for carbon dioxide adsorption [11,12,13,14,15,16,17,18,19,20,21]. Particularly, metal–organic framework (MOF)-based functional materials exhibited great adsorption capability for carbon dioxide due to their large surface area (even above 2000 m^2^·g^−1^), the similar pore size to the dynamic diameter of carbon dioxide molecules, and the interaction between carbon dioxide molecules and the imidazolium moieties on MOF-based compounds [22,23,24,25,26]. Thus, functionalization of MOF could be an effective way to design and synthesize solid sorbents for carbon dioxide adsorption.

Room-temperature ionic liquids (RTILs) are organic salts with a melting temperature below 100 °C. Typically, RTILs have high thermal/chemical stability, wide liquid range, low volatility, and good ion conduction, which are suitable characteristics for a wide range of applications [27,28,29]. It was reported that imidazolium-based RTILs exhibited excellent carbon dioxide adsorption capacity due to the high solubility of carbon dioxide in RTILs and the interaction of carbon dioxide molecules with the 2-position carbon on imidazole rings [30,31,32]. However, the high viscosity, particularly after carbon dioxide adsorption, makes the re-use of such sorbents quite difficult [33,34]. Hence, grafting or immobilization of imidazolium-based RTILs on porous solid support is expected to possibly address the viscosity-induced problems.

In this work, we report a new type of composite material containing zeolitic imidazolate framework (ZIF-67) and amine-functionalized imidazolium-based RTIL moieties, and we evaluate their carbon dioxide adsorption behavior at room temperature under relatively low pressure. It is proposed that the amino groups on imidazolium-based RTIL can be involved in the ZIF formation, leading to the imidazolium-type RTIL modified ZIF-67. The synthesized materials are expected to bear reasonably good carbon dioxide adsorption capacity due to the RTIL moieties and the large surface area of the synthesized molecules.

## 2. Experimental Section

### 2.1. Synthesis of 1-(3-Aminopropyl)-3-(2-bromoethyl)imidazole (NAIm-Br) 

To a solution of 1,2-dibromoethane (1.0 mL) in 5 mL of acetone, 1.0 mL of 3-aminopropyl imidazole was added under protection of nitrogen atmosphere. The mixture was then continuously stirred at room temperature for 24 h. The product was purified by distillation under vacuum to remove unreacted species. The final product of ionic liquid was a brown oil with yield of 92%, and the chemical structure (Figure 1a) was confirmed by spectroscopy. ^1^H NMR (D_2_O, 298 K, 300 MHz, ppm) δ: 1.81 (m, 2H, –N–C–CH_2_–C), 2.53 (t, 2H, N–CH_2_–C–C–N), 3.93 (m, 4H, N–CH_2_–C–C and Br–CH_2_–C), 4.17 (t, 2H, N–CH_2_–C–Br), 6.99 (S, 1H, N–CH=C), 7.04 (S, 1H, N–CH=C), 7.55 (s, 1H, N=CH–N). Fourier-transform infrared (FTIR) (KBr pallet, cm^−1^): 3419, 3094, 2930, 2508, 1628, 1598, 1564, 1510, 1488, 1381, 1260, 1205, 1148.

### 2.2. Synthesis of Ionic-Liquid-Modified ZIF-67

Firstly, 0.119 g of ionic liquid NAIm-Br was dissolved in 30 mL of methanol, and 1.232 g of 2-methylimidazole (Figure 1a) was added. After stirring the above mixture for 12 h at room temperature, 1.096 g of cobalt nitrate hexahydrate was added. The solution was kept for 24 h without stirring, and the purple precipitate was collected through centrifugation and extensive washing. The product was finally dried under vacuum at 80 °C. The obtained sample was not soluble in water and is denoted throughout as ZIF-67-IL.

### 2.3. Characterization

^1^H nuclear magnetic resonance (^1^H NMR) was performed to investigate the chemical structure of the synthesized ionic liquid on a Mercury VX-300 spectrometer (Varian, Beijing, China) using tetramethylsilane as the internal standard. Fourier-transform infrared (FTIR) spectra were recorded on a Nicolet 60SXB spectrometer (Thermo Fisher Scientific, MA, USA) with a resolution of 4 cm^−1^ to determine the synthesis of ionic liquids. Field-emission scanning electron microscopy (SEM) images were applied on JEM-7500F (JEOL, Tokyo, Japan) to characterize the morphology of ZIF-67 and the ionic-liquid-modified ZIF-67. Thermogravimetric analysis (TGA) and differential scan calorimetry (DSC) were performed using an STA449F3 thermal analyzer (Netzsch, Selb, Germany) under a dynamic heating mode with a ramp rate of 10 °C per minute in air. Nitrogen adsorption-desorption isotherms were applied to determine Brunauer-Emmett-Teller (BET) surface area and pore size distribution on a Micromeritics ASAP 2020 (Micromeritics, Norcross, GA, USA) instrument at 77 K. X-ray photoelectron spectroscopy (XPS, VG Multilab2000X, Thermo Fisher Scientific, MA, USA) was carried out to determine the near-surface elemental composition of ZIF-67 and ionic-liquid-modified ZIF-67. The carbon dioxide adsorption was performed on an ASAP 2020 gravimetric instrument within the pressure range of 0–800 mmHg. The adsorbed amount of carbon dioxide can be directly obtained from the change of the weight. The desorption of carbon dioxide for pre-adsorbed samples was carried out at 200 °C. Prior to the carbon dioxide measurements, samples were pre-heated to 200 °C and cooled down to room temperature under the protection of nitrogen atmosphere to avoid the influence of physically adsorbed water and carbon dioxide from air.

## 3. Results and Discussion

The motivation of this work relies on the modification of porous materials, ZIF-67 here, with imidazolium ionic-liquid moieties to enhance the adsorption capacity of carbon dioxide due to the high solubility and the interaction of carbon dioxide molecules with 2-position carbon atoms on imidazole rings. To this, the mixed 2-methylimidazole and synthesized ionic liquids were simultaneously used as ligands during the formation of ZIF-67. It was hypothesized that the amine groups (with lone-pair electrons) on the synthesized ionic liquid could also be involved in the formation of porous materials. To confirm this hypothesis, FTIR spectra were firstly recorded, as shown in Figure 1. The absorption band at about 426 cm^−1^ was assigned to Co–N vibration, indicating that the complexation between the nitrogen atom, which has lone-pair electrons, and cobalt ions, which have empty atomic orbitals, occurred for both ZIF-67 and ZIF-67-IL samples. Compared with the spectrum of ZIF-67, the newly appeared absorption bands at 1630 and 1281 cm^−1^ in the spectrum of ZIF-67-IL were attributed to the bending vibration of NH_2_ and the stretching vibration of N–C–H, qualitatively suggesting the incorporation of the synthesized ionic liquids with ZIF-67.

Figure 2 shows the XPS spectra of the synthesized composite of ZIF-67-IL. From the full XPS survey of ZIL-67-IL (Figure 2a), it can be concluded that the synthesized sample contained Co (781.16 eV), O (531.76 eV), N (398.97 eV), C (284.94 eV), and Br (68.26 eV), further confirming the successful formation of ZIF-67-IL. To further understand the bonding formation of ionic liquids and ZIF-67, the high-resolution spectra of Br, C, and N atoms were recorded. The high-resolution Br 3*d*5 spectrum (Figure 2b) can be deconvoluted into two peaks centered at 69.05 and 17.95 eV, suggesting the existence of ionic-liquid moieties. The high-resolution C 1*s* spectrum and its deconvolution result are displayed in Figure 2c. It can be seen that there were six different chemical environments for carbon atoms. The deconvoluted peaks of 282.95, 283.97, 284.75, 285.52, 286.15, and 186.70 eV corresponded to C=O, saturated C–C, C=C, C=N, C–N, and C–Br bonding, respectively. Figure 2d shows the high-resolution N 1*s* spectrum and the deconvolution results. Apart from the complexed 2-methylimidazole peaks at 398.32 and 399.00 eV for C–N and C=N [16], the peaks centered at 400.90 and 401.55 eV corresponded to the nitrogen atoms on the imidazole rings of the ionic liquid. The difference in binding energy of imidazole rings implied that the imidazole ring in 2-methyliimidazole had a strong interaction with cobalt ions, whereas the imidazole ring in the formed ionic liquids was not involved in the complexation. Moreover, the observed additional peak at 400.15 eV contributed to the complexation of amino groups with metal ions, suggesting that the amino groups contributed to the complexation and the according ZIF-67 formation.

Figure 3 shows the X-ray diffraction (XRD) patterns of ZIF-67 and ZIF-67-IL. It is apparent that the characteristic diffraction peaks of ZIF-67 remained after modification with imidazolium-based ionic liquids, indicating that the introduction of ionic liquids did not affect the crystallinity of ZIF-67 [35]. However, the decreased peak intensity indicates that the introduction of ionic liquids could lead to the electron density of nano-domains in ZIF-67. Moreover, SEM images of both ZIF-67 and ZIF-67-IL (Figure 4) revealed that both samples were dodecahedral crystals, indicating that only some of the ionic-liquid molecules were involved in the formation of ZIF-67, and the incorporated ionic liquids had little influence on the morphology of the formed products.

The surface area of the applied porous materials is critical for gas adsorption and separation. To determine the porous structure of ZIF-67-IL, nitrogen adsorption-desorption isotherms were recorded in Figure 5. It can be clearly seen that both ZIF-67 and ZIF-67-IL samples exhibited the characteristics of typical type I isotherms, indicating the existence of micropores. The accordingly calculated BET surface area, pore volume, and average pore size for both samples are listed in Table 1. It is apparent that the introduction of ionic liquids to ZIF-67 resulted in a slight decrease in surface area, pore volume, and average pore size, suggesting that the ionic liquids were located on the surface of ZIF through complexation between cobalt ions and amino groups on the synthesized ionic liquids.

Since the sample needs to be pretreated under high temperature to remove the adsorbed small molecules before performing carbon dioxide adsorption, the thermal stability of the synthesized samples was investigated, as shown in Figure 6. It can be seen that the pristine ZIF-67 was quite stable at temperatures below 250 °C, whereas the ZIF-67-IL started to decompose at around 220 °C, probably due to the decomposition of ionic-liquid moieties. Within the temperature range from 220 to 375 °C, the weight loss of the ZIF-67-IL was about 5.46 wt.%. Thus, the pretreated temperature prior to carbon dioxide adsorption was set to 200 °C in this work.

The carbon dioxide adsorption behavior was investigated by realizing the change of weight after adsorption of carbon dioxide at 25 °C or after desorption of pre-adsorbed samples at 200 °C under different pressures. The results are shown in Figure 7. It is apparent that the sample of ZIF-67-IL exhibited much higher adsorption capacity within the tested pressure ranges. For example, the adsorption capacity for ZIF-67-IL was about 24.9 cm^3^·g^−1^, whereas it was only about 16.3 cm^3^·g^−1^ for ZIF-67 under the pressure of 800 mmHg. Moreover, the adsorbed amount calculated from the adsorption process was very similar to that calculated from the desorption process for both samples, indicating that the possible chemical adsorption was negligible for both samples. Thus, the improved carbon dioxide adsorption capacity of ZIF-67-IL compared to ZIF-67 was attributed to the enhanced solubility of carbon dioxide molecules in the ionic-liquid moieties considering the very similar porous structure of ZIF-67 and ZIF-67-IL. It should be noted that the adsorption capacity of the synthesized ZIF-67-IL under the tested conditions was very similar to reported solid sorbents in the literature [9,11,13,31,32]. However, the value is still not satisfactory due to the limitations of the testing technique. We do believe that, with the increase in the operation pressure, the carbon dioxide adsorption capacity under high pressure would be much higher than the value reported here.

## 4. Conclusion

An ionic-liquid-modified zeolitic imidazolate framework (ZIF-67) was synthesized using amine-functionalized imidazolium-based ionic liquids and 2-methylimidazole as co-ligands during the formation of ZIF. The introduction of ionic liquid moieties to ZIF-67 had a very slight influence on the surface area and porous structure of ZIF-67. The complexation between the lone-pair electrons of N-atoms on amino groups in the ionic liquid and cobalt ions resulted in the grafting of ionic-liquid moieties on the surface of ZIF-67. Due to the high solubility of carbon dioxide molecules in the ionic-liquid moieties, the carbon dioxide adsorption capacity of ionic-liquid-modified ZIF-67 was much higher than that of pristine ZIF-67 within the tested pressure ranges. The presented results demonstrate that the modification of porous materials with ionic liquids can improve their carbon dioxide adsorption behavior.

## Figures and Tables

**Figure 1 materials-12-02361-f001:**
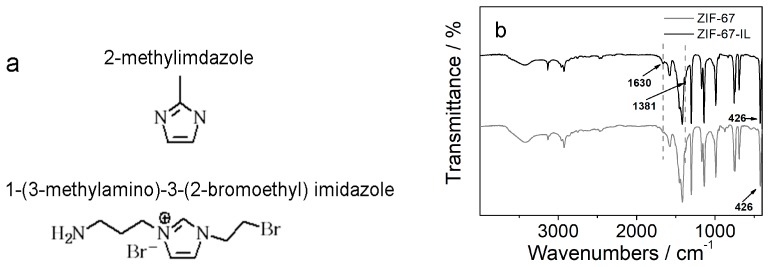
(**a**) The chemical structure of precursors used for formation of ionic-liquid-modified zeolitic imidazolate framework (ZIF-67-IL). (**b**) Fourier-transform infrared (FTIR) spectra of ZIF-67 (gray line) and ZIF-67-IL (black line). The dashed lines are guides for the eyes.

**Figure 2 materials-12-02361-f002:**
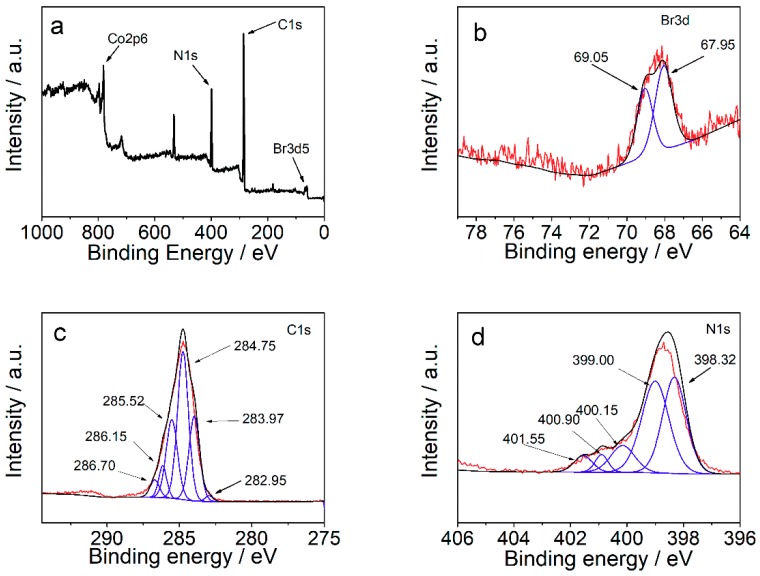
X-ray photoelectron spectroscopy (XPS) spectra of ZIF-67-IL: full survey (**a**), high-resolution spectrum and deconvolution results of Br 3*d*5 (**b**), C 1*s* (**c**), and N 1*s* (**d**).

**Figure 3 materials-12-02361-f003:**
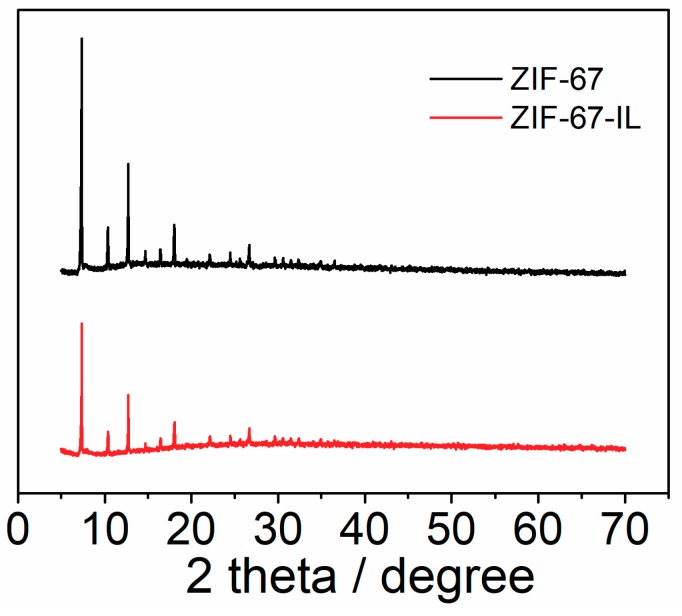
X-ray diffraction (XRD) patterns of ZIF-67 and ZIF-67-IL.

**Figure 4 materials-12-02361-f004:**
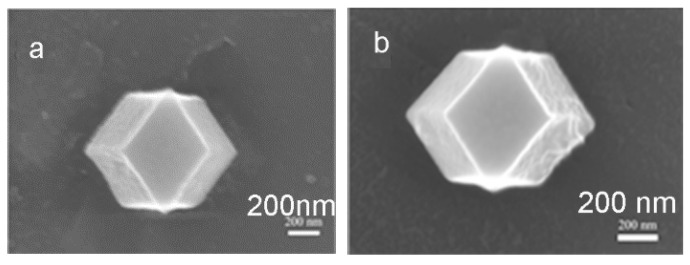
SEM images of ZIF-67 (**a**) and ZIF-67-IL (**b**).

**Figure 5 materials-12-02361-f005:**
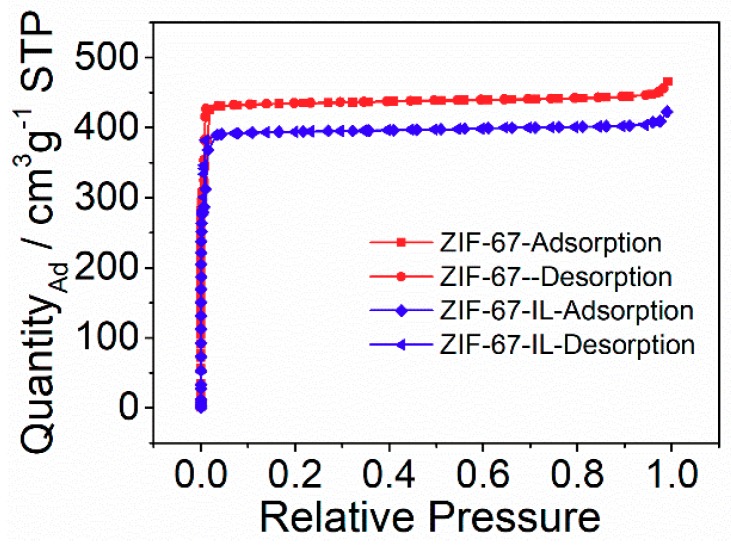
Nitrogen adsorption-desorption isotherms for ZIF-67 and ZIF-67-IL.

**Figure 6 materials-12-02361-f006:**
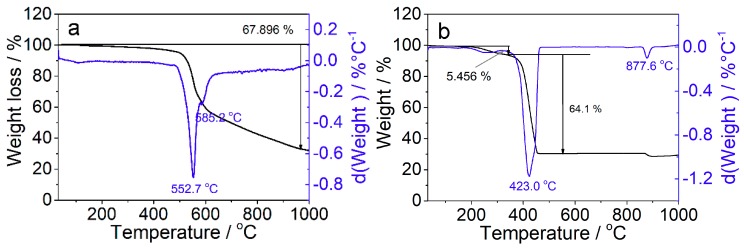
Thermogravimetric analysis (TGA) (black lines) and differential TGA curves (blue lines) of ZIF-67 (**a**) and ZIF-67-IL (**b**).

**Figure 7 materials-12-02361-f007:**
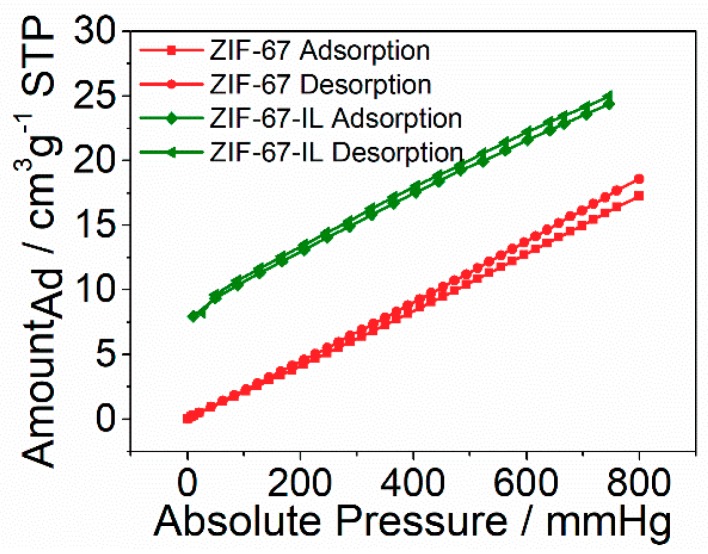
Carbon dioxide adsorption capability under different pressures at 25 °C for ZIF-67 and ZIF-67-IL.

**Table 1 materials-12-02361-t001:** Texture structure of the synthesized zeolitic imidazolate framework (ZIF-67) and ionic-liquid-modified ZIF-67 (ZIF-67-IL).

Samples	Surface Area (m^2^·g^−1^)	Pore Volume (cm^3^·g^−1^)	Pore Size (nm)
ZIF-67	1716	0.72	1.67
ZIF-67-IL	1707	0.65	1.53

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
