# Peer review of "Ionic Liquids-Functionalized Zeolitic Imidazolate Framework for Carbon Dioxide Adsorption"

_materials, 2019, doi:10.3390/ma12152361_

Round 1
Reviewer 1 Report
This manuscript is about the synthesis of ionic liquid functionalized zeolitic imidazolate frameworks (ZIFs) and their modifications, the latter serving as improved CO2adsorbent. This topic has a broad interest in the reduction of global warming gasses and environmental science. The authors use experimental probes, such as nuclear magnetic resonance, FTIR, field emission scanning electron microscopy, etc., to support their findings. I particularly like the correlation of the FTIR frequencies with bonding. It would be interesting in the future to perform computational simulations for cross-checking these results.
Overall, the manuscript is well written and contains the appropriate references. It should be accepted as is.
Minor improvements could be on figures axis, such as Figure 3, 5, and 7, where axis fonts seem unreasonably large. Typos should fixed. For example, line 153 page 5 a comma is needed before the word "where". Also line 160, page 6 a space is needed between "C" and "or " (i.e., replace "Cor" with "C or") and between "C" and "under" (i.e., replace "Cunder" with "C under").
Figure 6b, right y-axis should be blue.
Author Response
We thank the reviewer for evalutaion of the manuscript and the suggestion. We have modified the Figure 6b as the reviewer suggested and have carefully read through the manuscript. The typos were corrected in the revised manuscript as highlighted with red colors.

Reviewer 2 Report
I have reviewed the manuscript entitled:
„Ionic liquids-functionalized zeolitic imidazolate framework for carbon dioxide adsorption”.
In my opinion the manuscript need minor revision.
Global warning is related to the emission of carbon dioxide. Capture and storage of carbon dioxide is an important and interesting research topic.
Comment1
Line 57. The structural formula of 1-(3-aminopropyl)-3-(2-bromoethyl) imidazole should be added.
Comment 2
Line 66 The structural formula of 2-methylimidazole should be added.
Comment 3
Line 67. What is a cobalt oxidation state?
Comment 4
Line 67
The Cobalt salts can be toxic. Why authors using cobalt ions?
Comment 5
Is it possible to use other metals instated of cobalt?
Comment 6
What is the effect of cobalt on removal of carbon dioxide?
Comment 7
Line 69 The water is in the air. What is the solubility of ZIF-67 and ZIF-67-IL in water?
Comment 8
Authors should compare carbon dioxide adsorption capacity on ZIF-67 and ZIF-67-IL with other materials in the literature.
Comment 9
What is the advantage of ZIF-67 and ZIF-67-IL over other available materials for carbon dioxide removal?
Author Response
We thank the reviewer for the efforts made on our manuscript. We have carefully considered the comments and have made according changes in the revised maunscript as highlighted with red colors. The point-to-point response to the reviewer's comments were listed below:
Comment1
Line 57. The structural formula of 1-(3-aminopropyl)-3-(2-bromoethyl) imidazole should be added.
Comment 2
Line 66 The structural formula of 2-methylimidazole should be added.
Response: We thank the reviewer for the comment. The chemical structure of the mentioned two compounds were added in the revised manuscript as Figure 1a and also in the maintext.
Comment 3
Line 67. What is a cobalt oxidation state?
Response: In this work, we used cobalt nitrate as one of the precursors for the formation of ZIF-67 and the cobalt ion is divalent ions.
Comment 4
Line 67
The Cobalt salts can be toxic. Why authors using cobalt ions?
Response: We agree with the reviewer that the cobalt salt itself is toxic. However, in this work, the cobalt ions is traped within the formed framework through coordination with nitrogen atoms in imidazole and the toxcity should be significantly reduced. The idea to use cobalt ions is to form the framework with microporous structures.
Comment 5
Is it possible to use other metals instated of cobalt?
Response: Basically, the other transition metal ions with similar diameter for example iron or zinc ions should be possible to replace cobalt. However, the cobalt ions is so far the best choice for the formation of metal organic frameworks practically.
Comment 6
What is the effect of cobalt on removal of carbon dioxide?
Response: The initial idea to use cobalt is to form the framework which has large surface area and in turn is beneficial for carbon dioxide adsorption. However, we do think that the empty orbitals of cobalt ions is also helpful for adsorption of carbon dioxide through complexation.
Comment 7
Line 69 The water is in the air. What is the solubility of ZIF-67 and ZIF-67-IL in water?
Response: We apologize for the missed information. ZIF-67 and ZIF-67-IL are nanocrystals and are not soluble in water. We calrified this in the revised manuscript in Line 69 and Line 70 as highlighted with red colors.
Comment 8
Authors should compare carbon dioxide adsorption capacity on ZIF-67 and ZIF-67-IL with other materials in the literature.
Response: We thank the reviewer for the comment. We have added the according comments in Line 171 to Line 174 in the revised manuscript as highlighted with red colors.
Comment 9
What is the advantage of ZIF-67 and ZIF-67-IL over other available materials for carbon dioxide removal?
Response: We are sorry for the unclear description. The idea is to enhance the total carbon dioxide adsorption capacity by using the large surface area of ZIF framework for phyical adosprion and by using the amino group to enhance the chemical adsorption through reaction with carbon dioxide. We clarified this in the introduction part in Line 53 and Line 54 in the revised manuscript as highlighted with red colors.
